# Awareness and practice of medical waste management among healthcare providers in National Referral Hospital

Zimba Letho[1]*, Tshering Yangdon[2], Chhimi Lhamo[3], Chandra Bdr Limbu[3], Sonam Yoezer[3], Thinley Jamtsho[3], Puja Chhetri[3], Dawa Tshering[3]

1 Medical Education and Research Unit, Jigme Dorji Wangchuck National Referral Hospital, Thimphu, Bhutan, 2 Department of Pathology and Laboratory Medicine, Jigme Dorji Wangchuck National Referral Hospital, Thimphu, Bhutan, 3 Department of Nursing, Jigme Dorji Wangchuck National Referral Hospital, Thimphu, Bhutan

* zimba.letho@gmail.com

## Abstract

### Introduction

The management and treatment of Medical Waste (MW) are of great concern owing to its potential hazard to human health and the environment, particularly in developing countries. In Bhutan, although guidelines exist on the prevention and management of wastes, the implementation is still hampered by technological, economic, social difficulties and inade-quate training of staff responsible for handling these waste. The study aimed at assessing the awareness and practice of medical waste management among health care providers and support staff at the National Referral Hospital and its compliance with the existing National guidelines and policies.

### Materials and methods

An observational cross-sectional study was conducted from March to April 2019. Three research instruments were developed and used; (i) Demographic questionnaire, (ii) Aware-ness questions, and (iii) the Observational checklist. The data was coded and double entered into Epi data version 3.1 and SPSS version 18 was used for analysis. Descriptive statistics were used to present the findings of the study.

### Results

The majority of the respondents were female (54.1%) with a mean age of 32.2 (±7.67) years, most of whom have not received any waste management related training/education (56.8%). About 74.4% are aware of medical waste management and 98.2% are aware on the importance of using proper personal protective equipment. Only 37.6% knew about the maximum time limit for medical waste to be kept in hospital premises is 48 hours. About 61.3% of the observed units/wards/departments correctly segregated the waste in accor-dance to the national guidelines. However, half of the Hospital wastes are not being correctly transported based on correct segregation process with 58% of waste not segregated into infectious and general wastes.

**Data Availability Statement:** All relevant data are within the manuscript and its Supporting Information files.

**Funding:** The Jigme Dorji Wangchuck National Referral Hospital provided material support for this study but had no role in study design, data collection and analysis, decision to publish, or preparation of the manuscript.

## Conclusion

The awareness and practice of medical waste management among healthcare workers is often limited with inadequate sensitization and lack of proper implementation of the existing National guidelines at the study site. Therefore, timely and effective monitoring is required with regular training for healthcare workers and support staff. Furthermore, strengthening the waste management system at National Referral Hospital would provide beneficial impact in enhancing safety measures of patients.

## Introduction

A health care facility inevitably produces medical wastes (MW) that may be hazardous to health [1–3]. MW refers to all categories of waste generated from health facilities, clinics, animal husbandries, veterinary hospitals and other clinical laboratories, and home-based treatment of patients [3]. Although the MW is classified broadly, in general, it is categorized as Sharps, Infectious, Pathological, Pharmaceutical, cytotoxic, pressurized containers, chemical, radioactive and non-hazardous or general waste [1–4]. Hazardous MW is referred to any wastes which have the potential to cause harmful effects to human or environment if poorly managed [3]. Fig 1 provides the types of hazardous wastes, waste bin, and color code and biohazard symbols [5].

| Hazardous waste | PPE | Color-code | Bin description | Plastic bag | Symbols |
|---|---|---|---|---|---|
| Solid Infectious waste | Utility gloves, plastic apron, gumboot mask | *Red | **30L, strong leak-proof** plastic bin with **swing/pedal operated lid** and **wheels.** *place the plastic bag inside the waste bin)* | Biodegradable red plastic bag with **Biohazard symbol** |  |
| Sharps | Utility gloves, gumboot | *Yellow or *White | **Puncture proof sharps container/ boxes** with **biohazard symbol labeled as 'SHARPS'** | |  |
| Cytotoxic waste | Mask, goggle, Utility gloves, boot plastic apron & face shield | *Purple | Container or plastic bag with its symbol | |  |
| Chemical & Pharmaceutical waste | Utility gloves, plastic apron & mask,goggles, face shield | *Brown | Container or plastic bag | |  |
| Radioactive waste | Lead apron | | **Lead container** with **radioactive symbol**, labeled as "BIOHAZARD" | |  |

**Fig 1. Details of hazardous waste, PPE, color code and symbols of wastes.**

Although MW management practices differ from hospital to hospital, the implementation of MW management still poses a challenge. For the MW management, WHO has prepared various biomedical waste management guidelines to ensure the safe management of the wastes from healthcare facilities.

In Bhutan, the Government with various initiatives developed guidelines on proper management of healthcare waste including developing environmental code of practice for hazardous waste management 2002, waste prevention and management act of Bhutan 2009, Guideline for Infection Control and healthcare waste management in health facilities 2006 and National Infection Control and Medical Waste Management guidelines 2018 [3, 5, 6] however, implementation is still hampered by technological, economic, social difficulties and inadequate training of staff responsible for the handling of these waste [3, 7].

Jigme Dorji Wangchuck National Referral Hospital (JDWNRH), the apex Hospital of the capital city Thimphu, Bhutan caters to the healthcare services delivery for the population residing in Thimphu as well as to the patients referred from district hospitals. Being the only tertiary care hospital, large number of patients and referral samples are received at JDWNRH leading to increase medical waste generations. For the management of the waste generated, JDWNRH consist of infection control focal team headed by deputy nursing superintendent who supervises the overall monitoring of the infection control and waste management by conducting annual monitoring system in all the departments (Annual report 2018).

This study was conducted at JDWNRH which is also the teaching hospital affiliated to Khesar Gyelpo Medical University of Bhutan (KGUMSB). JDWNRH is 350 bedded hospital situated in Thimphu consisting of 1300 staff with an annual turnout of 5, 26,491 patients in OPD and 17,468 patients in IPD (Annual Report, 2017). This study targeted the staff and supporting staff working at National referral hospital on their awareness and practice on MW management practice.

The appropriate MW management process includes vital steps: Collection, segregation, storage, transportation, treatment, and disposal [2]. Table 1 show the procedure for collection, transportation, treatment of MW according to the National guideline [5].

Segregation is based on color-coding of the non-infectious and infectious wastes [4]. The segregated waste are stored in designated storage area within the units/wards/departments and storage duration is 24–48 hours (summer time) and 24–72 hours (winter time). The waste is collected every morning from the all the wards/unit/department and transported to the waste treatment unit. The infectious waste is treated by autoclaving and disposed off along with the

**Table 1. Collection, transportation, treatment and disposal of medical waste.**

| Hazardous waste | Collection | Transportation | Treatment Method | Disposal |
|---|---|---|---|---|
| Solid infectious waste | When **bin** is 3/4th full | **Only** on **specified waste trolley** or **cart** | 1. Autoclaving | Municipal bin |
| | | | 2. Chemical disinfection | Deep burial pit |
| | | | 3. Incineration | |
| Pathological waste | When **bin** is 3/4th full | | **Dispose** of in **deep burial pit** | Deep burial pit |
| Liquid infectious waste | Procedure specific collecting container | | Decontaminate with 0.5% bleaching solution in equal proportions (1:1) for 10 minutes | Sewage system with plenty of water |
| *Sharps | When the **box** is 3/4th full | | 1. Autoclaving & shredding or incineration | Deep burial pit or recycle |
| Chemical & Pharmaceutical waste | Collected and sent to pharmacy for final disposal | | Encapsulation | Landfill |
| Cytotoxic waste | Collect in leak-proof container and store in designated area | | 1. Encapsulation 2. Incineration 3. Chemical disinfection | Landfill and deep burial pit |
| Radioactive waste | Collect in lead container | | Decay by storage | |

general waste. The waste like cardboard box, pet bottles are sold to the vendor to reduce the waste going to the land fill and the food waste are used to prepare organic compost for fertilizer.

In an audit report by Royal Audit Authority (RAA) on Medical Waste Management conducted in 2008, it was found that the support staff handling MW of JDWNRH lacked awareness and knowledge on proper handling and management. Also, the waste handlers were seen handling the MW without protective gear such as utility gloves, apron, gumboots and mask [8]. Therefore, this study aimed at assessing the awareness and practice of health care providers on the management of medical wastes and implementation of the existing national guidelines.

## Materials and methods

An observational cross-sectional study was conducted to describe the awareness and current practice of medical waste management at JDWNRH. All Bhutanese citizenship health workers registered with Bhutan Medical and Health Council BMHC) and working permanently full-time in JDWNRH were interviewed including the supporting staff who handles the MW.

### Sampling

The convenience sampling method was used to collect data from all 18 departments of JDWNRH. The 18 departments were Clinical Departments (n = 15), Community Health Department (n = 1) and Diagnostic Departments (n = 2).

### Sample size

The sample size in this study was calculated using Krejcie and Morgan 1970 [9] formula to determine the sample size for finite population as follows:

$$S = \frac{X^2 NP(1 - P)}{d^2(N - 1) + X^2 P(1 - P)}$$

Where:
S = required sample size
$X$ = $Z$ value (e.g. 1.96 for 95% confidence level)
$N$ = population size
$P$ = population proportion [expressed as decimal, assumed to be .5 (50%)]
$d$ = degree of accuracy (5%), expressed as a proportion (.05); it is a marginal of error
In this study, 350 participants were recruited to target maximum number of participants from healthcare workers and supporting staff.

### Research instrument

Three research instruments were used in this study according to WHO standards (1) viz. i) Demographic questionnaire, (ii) Awareness questions, and (iii) the Observational checklist. All the research instruments were pilot tested and validated by the researchers prior to using on the participants.

### PART I: The demographic questionnaire

This part of the questionnaire was developed by the researcher which include all demographic variables (as given in the questionnaire) (S1 File).

## PART 2: Awareness questionnaire

This tool consists of 15 questions to test the awareness of medical waste management. Face to face interview was carried out to collect the data. The correct response was coded as 1 and incorrect as 0 respectively (S2 File).

## PART 3: Observational checklist

This checklist consist of assessing the process and practice on handling the MW by the healthcare workers and supporting staff by visual observation at the work station on the disposal method. The observation was coded as; 1 for Yes, 0 for No and 9 for Not applicable (S3 File).

## Data collection and analysis

Data were collected from March to April 2019. Head of Departments were explained on the objective of the study and the written informed consent were obtained from all the volunteer participants. Ethical clearance was sought from Research Ethic Board (REBH), Ministry of Health (Ref. No. REBH/PO/2019/012). Ethical waive was granted since there was no clinical intervention and the protocol fulfilled the criteria for ethical exemption from REBH.

The data was coded and double entered into Epi data version 3.1 and SPSS version 18 was used for analysis. Descriptive statistics were used to present the findings of the study. The current practice and awareness of medical waste management among health care providers in JDWNRH was described in terms of frequency, percentage, mean (*M*), and standard deviation (*SD*).

## Results

### Part 1: Demographic characteristics of participants

Table 2 shows the demographic characteristics of health care providers. A majority of the respondents were female (54.1%) as compared to males. The mean age of the health care providers was 32.2 years (SD = 7.35) with a minimum and maximum age of 20 and 55 years respectively. The age range of most health care providers was between 26 and 30 years (33.8%). Most of them (32.9%) had a Diploma followed by a Certificate degree (30.9%) and a bachelor's degree (18.8%). The average years of experience of the health care providers were 8.48 years (SD = 7.67) and 37.6% had an experience of fewer than 4 years. It also revealed that the highest respondents were nurses (44.1%) followed by technicians (23.5%) and least were health assistants (3.2%). Most of them had not received any waste management related training/education (56.8%). It is worth to note that 82.9% of health care providers have been vaccinated against Hepatitis B virus which is provided as mandatory vaccination for all healthcare providers and support staff.

### Part 2: Awareness questionnaire

Table 3 describes awareness about biomedical waste management among health care providers in JDWNRH. Almost all (98.5%) heard about medical waste and 69.7% are aware of regulation on medical waste management. About 74.4% of health care providers are aware of the biohazard symbol and only 45.3% knew about eight categories of medical waste.

It's encouraging to note that 83.5% and 88.2% of the respondents are aware that HIV/AIDS and Hepatitis B & C can be transmitted through medical waste respectively. Also, the majority (98.2%) are aware that personal protective measures are necessary while handling medical waste. 90.0% believe that the disinfection of medical waste is necessary and 72.9% are aware

**Table 2. Frequency and percentage of demographic characteristics** *(N = 340)*.

| Demographic characteristics | n | % |
|---|---|---|
| **Gender** | | |
| Male | 156 | 45.9 |
| Female | 184 | 54.1 |
| **Age in Years** (*m* = 32.2; *SD* = 7.35; *Min* = 20; *Max* = 59) | | |
| 20–25 years | 57 | 16.8 |
| 26–30 years | 115 | 33.8 |
| 31–35 years | 78 | 22.9 |
| >35 years | 90 | 26.5 |
| **Level of Education** | | |
| Certificate | 105 | 30.9 |
| Diploma | 112 | 32.9 |
| Bachelor's Degree | 64 | 18.8 |
| Higher than Bachelor's Degree | 19 | 5.6 |
| Others* (support staff minimum of 8th standard) | 40 | 11.8 |
| **Years of Experience** (*m* = 8.48; *SD* = 7.67; *Min* = 1; *Max* = 35) | | |
| 0–4 years | 128 | 37.6 |
| 5–8 years | 89 | 26.2 |
| 9–12 years | 45 | 13.2 |
| >12 years | 78 | 22.9 |
| **Place of Work** | | |
| Medical and Surgical (Oncology & Dialysis) | 84 | 24.7 |
| Emergency Department | 24 | 7.1 |
| Critical Care Units | 12 | 3.5 |
| Pediatrics | 09 | 2.6 |
| Community Health Department | 17 | 5.0 |
| Physiotherapy Department | 20 | 5.9 |
| Obstetrics/Gynecology | 20 | 5.9 |
| Laboratory Department | 25 | 7.4 |
| Pharmacy Department | 19 | 5.6 |
| Others (Orthopedic, Eye & ENT, Dental Department, OPDs) | 110 | 32.4 |
| **Profession** | | |
| Doctors | 25 | 7.4 |
| Nurses | 150 | 44.1 |
| Health Assistants | 11 | 3.2 |
| Technicians | 80 | 23.5 |
| Technologist | 26 | 7.6 |
| Others (Support staff) | 48 | 14.1 |
| **Training Attended on Waste Management** | | |
| Yes | 147 | 43.2 |
| No | 193 | 56.8 |
| **Vaccination against Hepatitis B** | | |
| Yes | 282 | 82.9 |
| No | 58 | 17.1 |

that a bleaching solution of 0.5% is used for the disinfection of infectious medical waste. However, only 37.6% are aware that the maximum time for medical waste to be kept in hospital premises is 48 hours.

**Table 3. Awareness about medical waste management among health care providers (N = 340).**

| Items | Aware | | Not Aware | |
|---|---|---|---|---|
| | n | % | n | % |
| 1. Have you ever heard about medical waste? | 335 | 98.5 | 5 | 1.5 |
| 2. Are you aware of regulation on medical waste management? | 237 | 69.7 | 103 | 30.3 |
| 3. Do you know about the biohazard symbol? | 253 | 74.4 | 87 | 25.6 |
| 4. Can you name eight categories of medical waste? | 154 | 45.3 | 186 | 54.7 |
| 5. Can you list down the guidelines provided for color coding in workplace? | 267 | 78.5 | 73 | 21.5 |
| 6. What is puncture proof container for sharps? | 265 | 77.9 | 75 | 22.1 |
| 7. What is the correct bag for disposal of cytotoxic drugs? | 73 | 21.5 | 267 | 78.5 |
| 8. What is the correct bag for intravenous sets, catheters and tubes? | 263 | 77.4 | 77 | 22.6 |
| 9. HIV AIDS can be transmitted through medical waste. | 284 | 83.5 | 56 | 16.5 |
| 10. Hepatitis B and C can be transmitted through medical waste. | 300 | 88.2 | 40 | 11.8 |
| 11. Personal protective measures are necessary while handling medical waste. | 334 | 98.2 | 6 | 1.8 |
| 12. When do you discard medical waste from the bin? | 196 | 57.6 | 144 | 42.4 |
| 13. Do you know about the methods for medical waste treatment? | 152 | 44.7 | 188 | 55.3 |
| 14. Disinfection of medical waste is necessary. | 306 | 90.0 | 34 | 10.0 |
| 15. Bleaching solution of 0.5% is used for the disinfection of infectious medical waste. | 248 | 72.9 | 92 | 27.1 |
| 16. The maximum time for medical waste to be kept in the hospital premises is 48 hours. | 128 | 37.6 | 212 | 62.4 |

## Part 3: Observational checklist

Table 4 describes the observation of the current practice on medical waste management for four categories; a) condition of waste receptacles, b). Segregation of waste, c) Transportation of medical waste, d) Appropriate use of PPE. The observation was carried out by visiting the department/unit/wards and visually observing whether the waste disposal process was followed in accordance to National guidelines on infection control (3).

On an average, 93.5% of the waste bins were appropriately available in the required color-coded bins (83.85%), however, the availability of blue-colored waste bin was minimum (45.2%). Only 58.1% of the waste bins were covered with 74.2% being foot-operated. The biohazard symbol was imprinted in the majority of the waste bin (90.3%) with 67.7% user posters displayed in waste bins.

About 61.3% of the observed units/wards/departments have correctly segregated the waste accordingly. Only 48% of the waste generated is transported in accordance with the transportation guideline with 58% of the waste not segregated into infectious and general wastes. Only 35.4% was found to be using appropriate PPE with 32.3% not complying and 32.3% not applicable.

## Discussion

The majority of the respondent were nurses which are concurrent with the highest number of health workers at JDWNRH being nurses followed by paramedical staff. Of these, the respondents were mostly in the age group of 26–30 years old and having a diploma course certificate. Less than half of the health care workers (43.2%) attended training on medical waste management which was a similar finding from the study conducted in 2015 [10]. However, it is noteworthy that the majority of the health care workers were vaccinated against Hepatitis B virus which is mandatory for all healthcare providers and support staff. Although most of them were aware of the regulations on medical waste management, the failure to adhere to these guidelines may be due to a lack of inspection from the authorities and the absence of strict rules and

**Table 4. Observation of current practice of medical waste management.**

| Parameters | Yes | No | NA |
|---|---|---|---|
| | (%) | (%) | (%) |
| **A. Condition of waste receptacles** | | | |
| 1. Is green colored waste bin available in ward | 29 (93.5%) | 1 (3.2%) | 1 (3.2%) |
| 2. Is yellow colored waste bin available in ward | 28 (90.3%) | 3 (9.7%) | 0 |
| 3. Is red colored waste bin available in ward | 30 (96.8%) | 0 | 1 (3.2%) |
| 4. Is blue colored waste bin available in ward | 14 (45.2%) | 12 (38.7%) | 5 (16.1%) |
| 5. Has green bag been placed lining the inner side of green bin | 23 (74.2%) | 6 (19.4%) | 2 (6.5%) |
| 6. Has red bag been placed lining the inner side of red bin | 29 (93.5%) | 1 (3.2%) | 1 (3.2%) |
| 7. Has blue bag been placed lining the inner side of blue bin | 1 (3.2%) | 20 (64.5%) | 10 (32.3%) |
| 8. Is green bag securely fitted with the bin | 23 (74.2%) | 6 (19.4%) | 2 (6.5%) |
| 9. Is red bag securely fitted with the bin | 27 (87.1%) | 3 (9.7%) | 1 (3.2%) |
| 10. Is blue bag securely fitted with the bin | 7 (22.6%) | 15 (48.4%) | 9 (29.0%) |
| 11. Are waste bins covered | 18 (58.1%) | 12 (38.7%) | 1 (3.2%) |
| 12. If covered, is cover foot-operated | 23 (74.2%) | 6 (19.4%) | 2 (6.5%) |
| 13. Is the biohazard symbol imprinted over waste bags | 28 (90.3%) | 3 (9.7%) | 0 |
| 14. Are posters to guide users displayed near waste bins | 21 (67.7%) | 10 (32.3%) | 0 |
| **B. Segregation of waste** | | | |
| 15. Does green bag contain only general waste | 25 (80.6%) | 3 (9.7%) | 3 (9.7%) |
| 16. Does yellow bag contain only sharp waste | 27 (87.1%) | 1 (3.2%) | 3 (9.7%) |
| 17. Does red bag contain only soiled infected waste | 21 (67.7%) | 8 (25.8%) | 2 (6.5%) |
| 18. Does blue bag contain only food waste | 3 (9.7%) | 12 (38.7%) | 16 (19.4%) |
| **C. Transportation of medical waste** | | | |
| 19. Appropriate on-site transport of medical waste used | 18 (58.1%) | 4 (12.9%) | 9 (29.0%) |
| 20. Is transportation of medical waste done during non-busy hours | 15 (48.4%) | 16 (51.6%) | 0 |
| 21. Are infectious and general waste transported separately | 13 (41.9%) | 18 (58.1%) | 0 |
| **D. Appropriate use of PPE** | 11 (35.4%) | 10 (32.3%) | 10 (32.3%) |

regulations. Therefore framing rules and regulations followed by proper and timely reminders of the importance of adhering to the rules and regulations are important by hospital infection control team.

There was satisfactory knowledge of color coding of wastes which is an essential factor for proper segregation of waste (80%) which was similar to the study conducted in Nigeria with 81.9% [11]. Our study revealed that the majority of the waste bins were color-coded (83.85%) which indicates the understanding of the respondents on the management of medical waste into infectious and non-infectious waste.

The waste generated is required to be transported by following the national guideline [3, 5]. Such wastes are collected and transported using a trolley, wheeled barrow, trucks, etc. Data from this study revealed that the waste is transported in trolleys and supporting staff loads it into the trucks by hand which could be dangerous. Although WHO stipulates that different trolleys should be used in transporting the different categories of wastes, this requirement does not adhere to 58% of wastes not segregated at source. Indeed, all the wastes generated are carried with the same trolley and this could also lead to cross-contamination [12].

As important as protective equipment is to anybody who handles medical wastes, only 35.4% complied with the use of appropriate PPE which is not consistent with the recommended standard of WHO which requires the use of heavy-duty gloves, boots, and apron [11]. There is a need to properly equipped and educate those in charge of on-site transportation of wastes, given the great danger associated with this task.

## Conclusion

Although national regulations exist on medical waste management, adherence to the practice is often limited due to inadequate sensitization amongst the health care workers and support staff. Most notably, the use of appropriate PPE while handling waste is often neglected causing potential risk. Therefore, timely and effective monitoring from the authorities should be implemented and regular training sessions to be provided for the healthcare workers and support staff.

Our findings suggest that although many of the respondents are aware of the National guideline on infection control and waste management, the practice is limited due to lack of proper training and sensitization on medical waste management process. Even though the hospital infection control team performs the assessment on waste management bi-annually, the findings and corrective action implementation needs to be strengthened by regular follow-up on action plan and providing incentives for the best medical management practice in the units/wards/departments.

## Supporting information

**S1 File. Demographic questionnaire.**
(DOCX)

**S2 File. Awareness questions.**
(DOCX)

**S3 File. Observational checklist.**
(DOCX)

## Acknowledgments

Authors would like to thank the Department of Nursing and JDWNRH administration and all the health care provides who were involved in the study and provided their immense support.

## Author Contributions

**Conceptualization:** Zimba Letho, Tshering Yangdon, Chhimi Lhamo.

**Data curation:** Zimba Letho, Chandra Bdr Limbu, Sonam Yoezer, Thinley Jamtsho, Puja Chhetri, Dawa Tshering.

**Formal analysis:** Zimba Letho, Chandra Bdr Limbu, Sonam Yoezer, Thinley Jamtsho, Puja Chhetri, Dawa Tshering.

**Funding acquisition:** Zimba Letho, Chhimi Lhamo.

**Investigation:** Zimba Letho, Tshering Yangdon, Chhimi Lhamo, Chandra Bdr Limbu, Sonam Yoezer, Thinley Jamtsho, Puja Chhetri.

**Methodology:** Zimba Letho, Tshering Yangdon, Chandra Bdr Limbu, Sonam Yoezer, Thinley Jamtsho, Puja Chhetri.

**Software:** Zimba Letho.

**Supervision:** Zimba Letho, Tshering Yangdon.

**Validation:** Tshering Yangdon.

**Writing – original draft:** Zimba Letho, Tshering Yangdon.

**Writing – review & editing:** Tshering Yangdon.

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
