## [Decision Letter · Decision Letter 0]

7 May 2020

PONE-D-20-06712

Awareness and Practice of Biomedical Waste Management among Healthcare Providers In

National Referral Hospital

PLOS ONE

Dear Ms yangden,

Thank you for submitting your manuscript to PLOS ONE. After careful consideration, we feel that it has merit but does not fully meet PLOS ONE’s publication criteria as it currently stands. Therefore, we invite you to submit a revised version of the manuscript that addresses the points raised during the review process.

We would appreciate receiving your revised manuscript by Jun 21 2020 11:59PM. To enhance the reproducibility of your results, we recommend that if applicable you deposit your laboratory protocols in protocols.io, where a protocol can be assigned its own identifier (DOI) such that it can be cited independently in the future. For instructions see: http://journals.plos.org/plosone/s/submission-guidelines#loc-laboratory-protocols

We look forward to receiving your revised manuscript.

Kind regards,

Itamar Ashkenazi

Academic Editor

PLOS ONE

Journal Requirements:

"Ethical clearance was sought from Research Ethics Board of Bhutan [REBH], Ministry of Health and JDWNRH management. Anonymity of the participants was ensured by not entering their names or any other identifiable information and a code was assigned to each case. The data collected was kept confidential and accessible only to the researcher. The findings was reported as a group and not as individual".  

Please amend your current ethics statement to confirm that your named institutional review board or ethics committee specifically approved this study.

Once you have amended this statement in the Methods section of the manuscript, please add the same text to the “Ethics Statement” field of the submission form (via “Edit Submission”).

For additional information about PLOS ONE ethical requirements for human subjects research, please refer to " ext-link-type="uri" xlink:type="simple">http://journals.plos.org/plosone/s/submission-guidelines#loc-human-subjects-research."

3. Please provide additional details regarding participant consent. In the ethics statement in the Methods and online submission information, please ensure that you have specified (a) whether consent was informed and (b) what type you obtained (for instance, written or verbal). If your study included minors, state whether you obtained consent from parents or guardians. If the need for consent was waived by the ethics committee, please include this information.”

4. Please consider modifying your title to ensure that it is specific, descriptive, concise, and comprehensible to readers outside the field (for example by  including the name of the centre/location where you carried out this cross-sectional study)

"Fund was supported from JDWNRH"

"The funders had no role in study design, data collection and analysis, decision to

publish, or preparation of the manuscript"

6. Please amend the manuscript submission data (via Edit Submission) to include author Chhimi Lhamo, Chandra Bdr Limbu, Sonam Yoezer, Thinley Jamtsho, Puja Chhetri and Dawa Tshering.

7. PLOS requires an ORCID iD for the corresponding author in Editorial Manager on papers submitted after December 6th, 2016. Please ensure that you have an ORCID iD and that it is validated in Editorial Manager. To do this, go to ‘Update my Information’ (in the upper left-hand corner of the main menu), and click on the Fetch/Validate link next to the ORCID field. This will take you to the ORCID site and allow you to create a new iD or authenticate a pre-existing iD in Editorial Manager. Please see the following video for instructions on linking an ORCID iD to your Editorial Manager account: https://www.youtube.com/watch?v=_xcclfuvtxQ

Reviewers' comments:

Reviewer's Responses to Questions

**Comments to the Author**

1. Is the manuscript technically sound, and do the data support the conclusions?

Reviewer #1: Partly

Reviewer #2: Partly

2. Has the statistical analysis been performed appropriately and rigorously? 

Reviewer #1: I Don't Know

Reviewer #2: No

3. Have the authors made all data underlying the findings in their manuscript fully available?

Reviewer #1: Yes

Reviewer #2: Yes

4. Is the manuscript presented in an intelligible fashion and written in standard English?

Reviewer #1: No

Reviewer #2: Yes

5. Review Comments to the Author

Reviewer #1: Review and Recommendations PONE-D-20-06712: Awareness and Practice of Biomedical Waste Management among Healthcare Providers in National Referral Hospital

Research Summary:

The authors conducted an observational cross-sectional study to evaluate healthcare provider awareness, knowledge, and practice concerning medical waste handling at a national referral hospital. The study sought to assess the familiarity with- and practices used by a variety of healthcare workers in their current work at a large medical facility as it pertains to their daily work assignments involving the handling of medical waste generated during patient care. While study participants were aware of potential risks associated with handling and disposing of medical waste, only a small proportion of the them followed best waste management practices. National and international guidelines exist regarding the proper handling of medical waste and the identification of potential health risks to healthcare providers improper handling presents. Deficiencies contributing to challenges encountered in the proper handling and full implementation of a compliant medical waste handling program are identified.

The identification of specifics regarding deficiencies and discussion of their potential impact is limited. The manuscript would be enhanced by providing additional descriptive information about training programs, facility enforcement measures, the range of available personal protective equipment and its use by the healthcare providers as well as what program resources are dedicated to medical waste program activities. Background information introduced in different sections of the manuscript could be pulled together more completely earlier in the document. Additional description in the methods of the rationale for approaches used in this study would enhance understanding of how the authors selected the evaluation tools used. This would add additional insight into any challenges investigators encountered. Including the strengths and limitations of the study and the methods used will provide other investigators with additional considerations for further research efforts involving medical waste handling programs in healthcare settings. Inclusion of some overall findings from this study to support the authors’ conclusion and recommendations would provide readers seeking considerations for improving medical waste handling in healthcare settings and the protection of healthcare workers from potential hazards associated with medical waste is suggested.

General Reviewer comments:

From my review of the manuscript it appears that the investigators’ presentation of their findings would benefit from some additional revision. The current organization of the manuscript along with identified information gaps suggest additional work is needed to refine this work prior to publication and reflect more fully the extensive work that was done. More in-depth examination of the data collected along with additional background and more extensive discussion of their findings would help develop their findings and the potential value of this work for other researchers and readers interested in this topic. While all of the data (responses) from the study participants that is presented in the manuscript appears to be found in the tables, I think the results and their presentation as well as the discussion and conclusions would be more informative if additional findings extracted from the collected data across respondents would have been included. Greater characterization within the subgroups of their attributes and then comparison with the other groups within the larger category could provide additional insights strengthening your findings. More specific comments and questions are provided throughout this review. The authors may want to consider whether incorporating some graphical presentation of their results would help with conveying the results rather than leaving it only to some large tables backing up the written results presentation. Were any photos obtained as part of the observational checklist process? That could add to the presentation of the findings. The manuscript organization along with the varying use of subheadings became increasingly confusing from lines 102 – 199. Please spell out acronyms the first time they are used.

The complete review document is found in the attached Word document. Total character count exceeded 22,000 character limit.

Reviewer #2: The manuscript titled “Awareness and Practice of Biomedical Waste Management among Healthcare Providers In National Referral Hospital” presents the results of the questionnaire study conducted among health care providers related to the medical waste management in the National Referral Hospital in Bhutan. The subject of the study is consistent with the thematic scope of the journal, but although is very important for national public health in Bhutan, it does not contain elements of novelty on a global scale. The study was well planed and correctly conducted but the description of the Methods and materials section is too general, the lack of some essential information. Results are presented by absolute values and percentages without analyses of selected correlations e.g. between awareness and job seniority, position or education level and statistical significance indicators. Additionally the presented data should be grouping in thematic issues instead of the showing the raw questions from the questionnaires (tables 2 and 3). Discussion section is too general and includes repetitions of details from Methods and Results sections. However, it is worth noting that the language of the manuscript is generally written in standard English and conclusions are supported by the presented data.

Minor comments:

- the key words could be extended to improve the visibility of the article;

- lines 27-28: not clear sentence

- lines 29-30: not clear sentence

- Methods and materials sections: Lack of very important details about the process of obtaining consent from responders to participate in the study (e.g. how they were asked, in writing or orally; with the information about the study of the hospital management or supervisor or without such consent; whether the respondents were informed about the agreement between the authors and the superiors).

- There is not clear who filled in the Observational Checklists and how numerous was the group of responders.

- line 129: “15 questionnaires”?

- lines 160-162: not clear sentence

6. PLOS authors have the option to publish the peer review history of their article (what does this mean?). If published, this will include your full peer review and any attached files.

Reviewer #1: No

Reviewer #2: No

---

## [Author Response · Author response to Decision Letter 0]

3 Aug 2020

1. Please ensure that your manuscript meets PLOS ONE's style requirements, including those for file naming. The PLOS ONE style templates can be found

We have edited and followed the guidelines of style/ template in according to PLOS ONE 

"Ethical clearance was sought from Research Ethics Board of Bhutan [REBH], Ministry of Health and JDWNRH management. Anonymity of the participants was ensured by not entering their names or any other identifiable information and a code was assigned to each case. The data collected was kept confidential and accessible only to the researcher. The findings was reported as a group and not as individual". 

Please amend your current ethics statement to confirm that your named institutional review board or ethics committee specifically approved this study.

Once you have amended this statement in the Methods section of the manuscript, please add the same text to the “Ethics Statement” field of the submission form (via “Edit Submission”).

For additional information about PLOS ONE ethical requirements for human subjects research, please refer to http://journals.plos.org/plosone/s/submission-guidelines#loc-human-subjects-research."

We have attached the ethical waive from the REBH in supporting document. 

3. Please provide additional details regarding participant consent. In the ethics statement in the Methods and online submission information, please ensure that you have specified (a) whether consent was informed and (b) what type you obtained (for instance, written or verbal). If your study included minors, state whether you obtained consent from parents or guardians. If the need for consent was waived by the ethics committee, please include this information.”

We have attached the consent form for reference

---

## [Decision Letter · Decision Letter 1]

24 Sep 2020

PONE-D-20-06712R1

Awareness and Practice of Medical Waste Management among Healthcare Providers in National Referral Hospital

PLOS ONE

Dear Dr. yangden,

Thank you for submitting your manuscript to PLOS ONE. After careful consideration, we feel that it has merit but does not fully meet PLOS ONE’s publication criteria as it currently stands. Therefore, we invite you to submit a revised version of the manuscript that addresses the points raised during the review process.

Being the academic editor reponsible for this submission, I decided to let the readers decide on the strengths and weaknesses of this manuscript. However, some concerns still need to be addressed:

1) As noted by Reviewer #1,  some clarifications, which were included in your Response to Reviewer, have not been reported in the revised  main text (for example, the sampling method choice has been justified in the Response to reviewers, but not in the main text; please address this concern in the manuscript text)

2) Please edit the manuscript carefully. At present, we note that the numbering of the Tables is incorrect (for example, two different tables are labelled "Table 1"); and that some of the references are incorrectly displayed , as highlighted by Reviewer #1. 

We look forward to receiving your revised manuscript.

Kind regards,

Carmen Melatti

Associate Editor

PLOS ONE

on behalf of 

Itamar Ashkenazi

Academic Editor

PLOS ONE

Reviewers' comments:

Reviewer's Responses to Questions

**Comments to the Author**

1. If the authors have adequately addressed your comments raised in a previous round of review and you feel that this manuscript is now acceptable for publication, you may indicate that here to bypass the “Comments to the Author” section, enter your conflict of interest statement in the “Confidential to Editor” section, and submit your "Accept" recommendation.

Reviewer #1: (No Response)

Reviewer #2: (No Response)

2. Is the manuscript technically sound, and do the data support the conclusions?

Reviewer #1: Partly

Reviewer #2: Yes

3. Has the statistical analysis been performed appropriately and rigorously? 

Reviewer #1: I Don't Know

Reviewer #2: Yes

4. Have the authors made all data underlying the findings in their manuscript fully available?

Reviewer #1: Yes

Reviewer #2: Yes

5. Is the manuscript presented in an intelligible fashion and written in standard English?

Reviewer #1: Yes

Reviewer #2: Yes

6. Review Comments to the Author

Reviewer #1: Review PONE-D-20-06712R1: Awareness and Practice of Biomedical Waste Management among Healthcare Providers in National Referral Hospital

Research Summary:

The authors conducted an observational cross-sectional study to evaluate healthcare provider awareness, knowledge, and practice concerning medical waste handling at a national referral hospital. The study sought to assess the familiarity with- and practices used by- a variety of healthcare workers in their current work at a large medical facility as it pertains to their daily work assignments involving the handling of medical waste generated during patient care. While study participants were aware of potential risks associated with handling and disposing of medical waste, only a small proportion of the them followed best waste management practices. National and international guidelines exist regarding the proper handling of medical waste and the identification of potential health risks to healthcare providers improper handling presents. Deficiencies contributing to challenges encountered in the proper handling and full implementation of a compliant medical waste handling program are identified.

The identification and development of specifics regarding findings from the study among the participants involving medical waste and discussion of their potential impact remains limited. Some text changes are noted in the revised manuscript in response to review comments but actual clarification addressing the original comments and questions was difficult to find. Added clarifying and supporting information in response to reviewer comments has primarily been included through the addition of two new tables and revision of headings and subheadings for sections within the manuscript to improve readability. Some minor changes in the text are also noted. Overall manuscript changes and clarification or the provision of rationale as to why addressing reviewer comments is not needed because of information already presented in the manuscript by the authors appears minimal.

General Reviewer comments:

From my review of the initial manuscript it appeared that the investigators’ presentation of their findings would benefit from additional revision more fully responding to the original review comments. Changes identified in the revised manuscript along with responses to review comments are limited for a major revision. Additional problems appear in the revised document – some immediate ones are identified in the following section. Overall, I think the revisions made and responses to reviewer comments appear to be very limited or unresponsive.

Specific Comments:

Review comment responses of “Changes made and rectified” are nonspecific as to what revisions the authors made.

Requested clarifications and information to strengthen the original manuscript reporting the study, its conduct, results, and their significance is limited in this revision.

The track-changes manuscript copy including the deletions and revisions made by the authors in response to review comments apparently was unavailable

The presence and discussion of two Table 1’s and two Table 2’s in the revised manuscript indicates an immediate problem concerning the revision process.

Additional information suggested for inclusion such as identifying 18 departments included in the study were listed in the author response to reviewer comments. Within the revised manuscript (lines 120-121) presentation of this information is minimal.

Deletions in the revision required a line-by-line comparison of the original manuscript with the revised version along with in-hand reference to the original review comments and the authors’ responses to identify actual changes. It appears from this process that not all comments were addressed.

Some reference changes and deletions were noted along with missing references for works cited in the manuscript (line 86 - Annual Report 2017; line 123 - Krejcie and Morgan 1970). Other examples of reference problems include: reference 11 line 243 now identifying a descriptive study in Mosul Nurs J 2013 which, while the text is unchanged from the original manuscript, originally cited the National Integrated Solid Waste Management Strategy, Thimphu, Butan 2014; and reference 10 line 247 originally cited a 1999 WHO document but now cites a 2015 article from the Journal of Environmental Management. In this second example the manuscript text also remains unchanged.

The revision of the manuscript and response of the authors to reviewer comments is incomplete.

Reviewer #2: Text is well prepared and fulfills all requirements for scientific work. Manuscript can be accepted to publication in scientific journal.

7. PLOS authors have the option to publish the peer review history of their article (what does this mean?). If published, this will include your full peer review and any attached files.

Reviewer #1: No

Reviewer #2: No

---

## [Author Response · Author response to Decision Letter 1]

23 Nov 2020

1. Changes made in the main text under subheading 'sampling' 

2. Changes made and table labelled accordingly 

Response to Reviewer #1: 

The tables mentioned is as per the content of the manuscript. For instance the tables used in the introduction part is mentioned as Table 1 and 2 respectively. Similarly, in the results part, different tables used for depicting the results are labeled as Table 1, 2 and 3 respectively. So there's appearance of two Table 1's and two table 2's based on its purpose on different headings. 

18 departments are segregated based on its purpose namely Clinicals, community health and diagnostic and not elaborated in order to reduce the use of space in the text. However, those 18 departments if need to be specified are as follows: 

1. Department of Anesthesiology and Critical Care

2. Department of Community Health

3. Department of Dentistry 

4. Department of Dermatology

5. Department of Emergency Medicine

6. Department of Forensic Medicine

7. Department of Medicine

8. Department of Obstetrics and Gynecology

9. Department of Ophthalmology

10. Department of ENT

11. Department of Orthopedics

12. Department of Pediatrics

13. Department of Pathology and laboratory Medicine

14. Department of Physiotherapy 

15. Department of Pharmacy 

16. Department of psychiatry 

17. Department of Radio Diagnosis and Imaging 

18. Department of Surgery

Reagarding ethical consideration, the Research Ethics Board of Health (REBH) under Ministry of Health is the ethical board responsible for the ethical considerations. link : http://www.moh.gov.bt/about/program-profiles/357-2/. Thus, the ethical clearance was sought from REBH prior to the start of the study and approved. The ethical statement has been presented only on the materials and method section of the manuscript.

---

## [Editor Report · Decision Letter 2]

27 Nov 2020

Awareness and Practice of Medical Waste Management among Healthcare Providers in National Referral Hospital

PONE-D-20-06712R2

Dear Dr. yangden,

We’re pleased to inform you that your manuscript has been judged scientifically suitable for publication and will be formally accepted for publication once it meets all outstanding technical requirements.

Kind regards,

Itamar Ashkenazi

Academic Editor

PLOS ONE
---

## [Editor Report · Acceptance letter]

11 Dec 2020

PONE-D-20-06712R2 

Awareness and Practice of Medical Waste Management among Healthcare Providers in National Referral Hospital 

Dear Dr. Yangdon:

I'm pleased to inform you that your manuscript has been deemed suitable for publication in PLOS ONE. Congratulations! Your manuscript is now with our production department. 

Kind regards, 

on behalf of

Dr. Itamar Ashkenazi 

Academic Editor

PLOS ONE